# Text-Free Federated Transformers Knowledge Distillation Without GAN

## Abstract

Federated Learning (FL) is a distributed learning process designed to protect user privacy by avoiding the transmission of user data during communication while training a model. Many techniques aim to enhance the performance of models through knowledge distillation but lack data on the server side. To address this issue, Generative Adversarial Networks (GANs) are commonly employed to generate data for model distillation. The GANs approach faces numerous challenges in recent popular large-scale Transformer-based NLP tasks, such as structural mismatches in models, high computational complexity, and concerns regarding the privacy of client-generated text. Prior research has sought to enhance the process using auxiliary data to avoid the above issues, however, the selection of suitable data tailored to diverse tasks remains a challenging endeavor. To address the challenges posed by GANs and auxiliary data, this work proposes a lightweight approach that samples from the embedding structure of Transformers and learns a set of pseudo data for the distillation process, which draws inspiration from the concept of soft prompts. This lightweight approach does not require GANs or auxiliary data, incurs no communication overhead, and yields improved model performance with relatively lower computational costs on the server side. Our experiments yield superior results compared to methods that rely on auxiliary data on complex NLP tasks such as the SuperGLUE Benchmark (Wang et al., 2019).

## 1 Introduction

Federated Learning (FL), a privacy-preserving distributed learning technique, has gained popularity amid growing concerns about data privacy and the rise of privacy protection laws in over 90 countries (Li et al., 2021). Applied in various fields, including Natural Language Processing (NLP) (Venkateswaran et al., 2022), Computer Vision (CV) (Lin et al., 2020), Industrial Artificial Intelligence (IAI) (Hao et al., 2019), and Medical Informatics (Xu et al., 2021), FL involves multiple clients collaboratively training a shared model. Two main categories include Cross-device for low-capacity clients like smartphones (Karimireddy et al., 2021) and Cross-silo for organizations with substantial computational resources (Huang et al., 2021). Noteworthy players such as Google (Bonawitz et al., 2019), Apple (Paulik et al., 2021), and Meta (Stojkovic et al., 2022) actively develop FL to ensure user privacy.

Data-free distillation, a crucial technical pathway to solve the data heterogeneity, i.e., the data in clients are non-identically and independently distributed (Non-IID) in federated learning, enhances the overall performance of the global model. Previous Data-free distillation of FL research like FedGEN (Zhang et al., 2022a) and FedFTG (Zhang et al., 2022a) based on GAN often relies on image tasks as primary benchmarks, neglecting NLP tasks. Other studies involving NLP tasks, such as FedAUX Sattler et al. (2021) and FedDF Lin et al. (2020), typically circumvent GANs or opt for auxiliary data distillation. Given the substantial gap in research on text tasks within FL's data-free distillation frameworks, exploring this area is both valuable and meaningful.

A practical issue arises with GAN in text generation (Alvarez-Melis et al., 2022). It is widely acknowledged that a notable obstacle in NLP is that GANs are unable to generate differentiable outputs, given the discrete nature of language models. This lack of differentiability hampers mainstream FL Data-Free distillation frameworks shown in Figure 1, such as FedGEN (Zhang et al., 2022a) and FedFTG (Zhu et al., 2021). These frameworks utilize GANs for target learning func-

tions but are ineffective in transmitting errors back to the generator, rendering them unsuitable for generating synthetic text in NLP FL distillation tasks.

Can alternative text generation models be employed aside from GANs? Our response is that the proposed solution entails substantial costs in terms of privacy preservation and computational overhead. The current state-of-the-art in text generation, such as large-scale Transformer models like GPT-4, necessitates self-supervised training, involving memorizing client-side text data. However, this introduces privacy concerns, requiring intricate training mechanisms (Ponomareva et al., 2022) and defense mechanisms (Guo et al., 2022) to ensure Transformer models remember the text while safeguarding privacy. Besides, memorizing client-side text data raises the following issues:

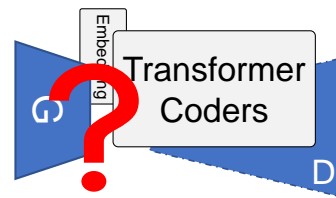

Figure 1: Creating a suitable and privacy-preserving generator for Transformers poses a formidable and intricate challenge.

- Concerning privacy complex training (Ponomareva et al., 2022) and defense mechanisms (Guo et al., 2022) are required to ensure the Transformer remembers the text while protecting privacy. Simultaneously, precautions must be taken to prevent attackers from reconstructing text through pre-trained models (Zhang et al., 2022b), introducing both computational and communication overhead.

- Regarding computational overhead deeper generative models (e.g., BERT, GPT) capable of memorizing more client-side text may exceed the computational capacity of the original task model, imposing a greater computational burden on clients.

- Concerning communication overhead, the baseline Transformer for the original task already consumes substantial communication, such as FedKD (Liu et al., 2019) using shallow Tiny-BERT resulting in 0.17-1.03GB per mobile client. Introducing language generation models like FedGEN would further amplify communication overhead.

**Our contributions** To tackle the distillation challenge in Federated Learning NLP tasks, particularly when dealing with Transformer models and limited auxiliary data, we present a text-free solution inspired by soft prompts. Our approach leverages diverse sampling from embeddings to effectively enhance model performance. We introduce three embedding sampling methods, each targeting improved distillation by optimizing samples through various objectives. This lightweight approach eliminates the need for GANs or auxiliary data, involves no communication overhead, and achieves enhanced model performance at relatively lower computational costs on the server side.

In cross-silo Federated Learning experiments on SuperGLUE benchmark tasks, using two downstream task models (with or without decoder structures), our approach outperforms solutions relying on auxiliary data. Ablation experiments highlight the unique advantages of models with embeddings, demonstrating the efficiency and quality of sampling in embedding-enhanced models.

## 2    RELATED WORKS

Federated learning faces several fundamental challenges such as imbalanced, non-iid data distribution and communication constraints. To solve these issues, we summarize two main approaches from previous works, namely the model optimization approach and the knowledge transfer approach.

**Model optimization** The model optimization approach, represented by neural network models, typically utilizes local optimization algorithms such as SGD and Adam on clients. The FedAvg (McMahan et al., 2017) algorithm, proposed with the concept of federated learning, is one of the most widely applied algorithms. Numerous studies have pointed out that the inconsistency between local and global optimization directions hinders achieving desirable results (Li et al., 2022). To address this issue, algorithms such as SCAFFOLD (Karimireddy et al., 2020) and FedOpt (Reddi et al., 2020) have been introduced, which incorporate regularization and local gradient corrections.

**Knowledge distillation** The knowledge distillation approach was originally used for model compression. FedDF (Lin et al., 2020) is the first algorithm to aggregate knowledge in FL using distillation techniques. FedDF uses GAN for image tasks and auxiliary data for NLP tasks. Later, in the domain of image generation tasks, the FedGEN (Zhu et al., 2021) algorithm employs GAN to learn

the local distribution and complement data on the server side. The FedFTG (Zhang et al., 2022a) algorithm utilizes GAN to learn difficult samples for the global model. Although these solutions have achieved good performance on classical image classification datasets, the instability of adversarial networks raises questions about their practical applicability. In the field of NLP, FedAUX (Sattler et al., 2021) is developed to enhance data distillation by leveraging classifier weights. However, the challenge of selecting appropriate auxiliary data for specific tasks still persists and knowledge distillation for generative models lacks auxiliary distillation schemes. Another distillation approach FedKD (Wu et al., 2022) involves distilling knowledge from a local large model to a global small model, effectively reducing communication cost while maintaining excellent performance. Recently, the distillation technique has evolved into the dataset condensation approach, which uses distillation techniques to compress data, such DOSF (Zhou et al., 2020) and FedDM (Xiong et al., 2022).

## 3 PRELIMINARIES

### 3.1 FEDERATED LEARNING

We consider the federated deep learning problem in cross-silo scenarios. There is a set of learning tasks $\mathcal{T} = \{T_1, T_2, \cdots, T_M\}$, and a dataset $D = \{(x, y)\}$, where data $(x, y)$ is from a distribution $\mathcal{D}$ and $x \in \mathcal{X}, y \in \mathcal{Y}$. To solve all of the tasks together, we aim to train a neural network as $f(x, w)$, and let $\hat{y} = f(x, \omega)$ represent the predicted label. The population loss of the training neural network parameters $\omega$ is $\mathcal{L}(\omega) = \mathbb{E}_{x \sim \mathcal{D}}[l(f(x, \omega), y)]$ . In the classification problems, we can take the loss function as the cross-entropy (CE) between the network output distribution and true distribution. For two discrete distributions $P$ and $Q$ with the same support $\mathcal{Y}$, their cross-entropy is defined as $\text{CE}(P||Q) = -\sum_{y \in \mathcal{Y}} P(y) \log Q(y)$.

For the cross-silo scenario, there are $K$ clients collectively working on the tasks. We abuse the notation 'clients' in this paper to denote the local servers with an input dataset. For each client, $k \in [K]$, the data of it is from the distribution $\mathcal{D}_k$, and this client $k$ could join in a subset of all tasks. All clients collaborate to obtain a global model $\omega$ with objective

$$\min_{\omega} \sum_{k=1}^{K} \mathbb{E}_{x \sim \mathcal{D}_k}[l(f(x, \omega), y)] . \tag{1}$$

**Knowledge Distillation** For KD in federated learning, typically it needs a proxy dataset $\mathcal{D}_P$ to minimize the discrepancy between the outputs from the teacher model $\omega_T$ and the student model $\omega_S$. A representative choice is to use Kullback-Leibler (KL) divergence to measure such discrepancy, it is defined as $\text{D}_{\text{KL}}(P||Q) = \sum_{y \in \mathcal{Y}} P(y) \log(\frac{P(y)}{Q(y)})$.

Consider in the neural network, let $f(\cdot)$ be the logits outputs and $\sigma(\cdot)$ be the softmax function. We can treat each client model $\omega_k$ as a teacher, then the information is aggregated into the student (global) model $\omega$ by:

$$\arg\min_{\omega_S} \mathbb{E}_{x \sim \mathcal{D}_P}[\text{D}_{\text{KL}}(\sigma(\frac{1}{K} \sum_{k=1}^{K} f(x, \omega_T))||\sigma(f(x, \omega_S)))] . \tag{2}$$

### 3.2 TRANSFORMERS

The parameter $\omega$ of a Transformer model consists of three main components: the word embedding layer $\omega^{emb}$, the encoder layers $\omega^{enc}$, and the optional decoder layers $\omega^{dec}$. These components are followed by a task-specific head $\omega^{lm}$, which outputs the corresponding labels for the given task $T$. The parameters except the embedding process are collectively referred to as the task parameters, we denote them as $\omega^f = \{\omega^{enc}, \omega^{dec}, \omega^{lm}\}$.

Two approaches can be considered for downstream tasks. The first approach is the classic discriminative Transformer. The final prediction probability of $x$ is obtained directly from the output at the [CLS] token position, which is embedded at the beginning of the input sentence. This can be represented as:

$$P(y|x,\omega) = \sigma(f(h(x;\omega^{emb});\omega^f)) \,, \tag{3}$$

where $h(\cdot)$ embeds $x$ into space $\mathcal{E}$ , $\sigma$ is the softmax function.

The second approach is the generative text-to-text model, which does not provide direct probabilities for the labels corresponding to the task. Instead, it generates a series of words corresponding to the task labels. The probability of the word sequence $q_{1:L}$ with input $x$ can be factorized as follows:

$$P(q_{1:L}|x,\omega) = \prod_{l=1}^{L} P(q_l|q_{1:L}, x, \omega) \,. \tag{4}$$

Here, $q_{1:L}$ represents the actual words corresponding to the predicted label $\hat{y}$. Typically, greedy search or beam search is used to determine the final word sequence $q_{1:L}$.

Overall, to maintain consistency in the output format of the model, whether it is a discriminative or generative model, the function for the discriminator of a Transformer-based model can be written $P(y|x,\omega) = \sigma(f(h(x;\omega^{emb}),\omega^f))$.

## 4 DIVERSITY RANDOMLY SAMPLE METHOD

This section will provide a more rational distillation objective and elucidate efficient methods for sampling embeddings. The previous distillation method, such as FedDF, weights all outputs of neural networks to obtain the teacher distribution. In contrast, our approach aims to fully consider the independent cross-silo by generating a new proxy dataset with both similar and different information among all clients.

As for the distinctive architecture of the Transformer model, it is not possible to distill the discrete embedding layer. Therefore, we directly obtain new embedding layer parameters of the student model $\omega_S$ as averaging $\omega_S^{emb} = \frac{1}{K} \sum_k^K \omega_k^{emb}$, and accomplish the distillation for other parameters $\omega_S^f$ with the objective

$$\arg\min_{\omega_S} \frac{1}{K} \sum_{k=1}^{K} \mathbb{E}_{\theta \sim \mathcal{D}_S}[\mathrm{D}_{\mathrm{KL}}(\sigma(f(\theta;\omega_k^f))||\sigma(f(\theta;\omega_S^f)))] \,. \tag{5}$$

Here, $\theta$ represents pseudo-embedding samples extracted from the proxy dataset $\mathcal{D}_S$ which will be constructed later. We demonstrate that only adjusting the parameters $\omega^f$ of the transformer on the server side is already effective. For some models like T5, we can keep the embedding layers fixed after the first time of training and not updated in the following stages to ensure the embedding layers of all clients are the same.

To achieve this goal, we need to design a scheme that allows for comprehensive sampling within the sample space of various client models. The simplest approach is to generate pseudo-samples by introducing noise that follows the same distribution as the data and embedding. However, randomly sampled noise may not necessarily lie within the sample space of client models. Inspired by (Ma et al., 2020), we adjust the noise parameters to align with the objective function of client models, thereby constructing effective pseudo-samples within the sample space of the clients.

**(1) Random sampling** In intuition, the input data for training the parts of the encoder and decoder is directly sourced from the embedding layer, so directly random sampling from the embedding seems like a reasonable operation. In practice, we have found that this random sampling method yields improvements in BERT models, but its effectiveness is limited in other models.

**(2) Target sampling** Randomly sampled data lacks purpose, making it challenging to guarantee its quality. Inspired by soft prompts, pseudo-samples extracted from the embeddings layer are subsequently optimized using the target loss to align with the distribution output by the teacher model on $\gamma_k^{tr}$. That is, we construct a target loss function by the cross-entropy as

$$\mathcal{L}_{tar} = \sum_k^K \mathrm{CE}(\sigma(f(\theta_k^{tr};\omega_k^f)); \gamma_k^{tr}) \,. \tag{6}$$

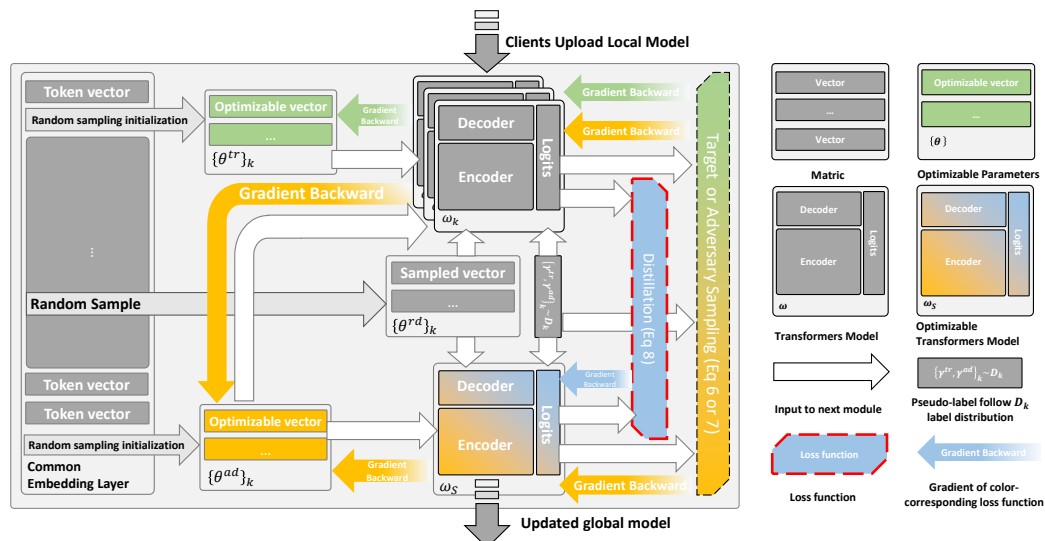

Figure 2: Logic flow of the Three Server-Side Sampling Methods in FedDRS. Following the upload of the model by the client, the process proceeds from left to right as follows: In the first stage, the forward phase (white arrows), samples three sets of initial sample parameters from the Embedding layer. Two of these samples are then fed into the model of clients and the global model. The loss for target sampling and adversarial sampling is computed (Eq 6 or 7). In the second stage, the backward phase (shaded arrows), noise samples are updated based on the distinct sampling losses. In the third stage, the distillation phase, the distributions derived from the three sets of samples are employed to distill the global model within the client model (Eq 8). The entire adversarial sampling process is iterated several times to obtain the most updated global model.

Here, $\gamma_k^{tr}$ represents a set of randomly generated pseudo-labels from distribution $\gamma_k$. Then, we will use the gradient descent method to optimize it. During this optimization process, we can get the pseudo samples.

**(3) Adversary sampling** In order to further enhance the quality of samples and increase the diversity of pseudo samples, we drew inspiration from the concept of FedFTG. Our objective is to obtain pseudo samples that exhibit correct $\gamma_k^{ad}$ labels on $\omega_k$ while incurring a significant loss on $\omega_S$. A sample that is correctly classified by the teacher model but misclassified by the student model can be considered as high-quality for the student, and therefore, it is deemed worth learning from.

$$\mathcal{L}_{adv} = \sum_k^K \text{CE}(\sigma(f(\theta_k^{ad}; \omega_k^f)); \gamma_k^{ad}) - \lambda \cdot \text{CE}(\sigma(f(\theta_k^{ad}; \omega_S^f)); \gamma_k^{ad}) . \tag{7}$$

Here, the parameter $\lambda$ controls the strength of the adversarial effect between the teachers (clients model) and student (global model), $\gamma_k^{ad}$ represents a set of randomly generated pseudo-labels from distribution $\gamma_k$. We can also use the gradient descent method to optimize this adversary loss function and get the pseudo samples.

**Distillation** Combining the aforementioned three sampling methods at the aggregate level results in a diversified set of pseudo-embeddings $D_p = \{\theta^{rd}, \theta^{tr}, \theta^{ad}\}_{k=1}^K$ and pseudo-labels $\{\gamma^{tr}\}_{k=1}^K, \{\gamma^{ad}\}_{k=1}^K$. This dataset can be employed to facilitate the distillation process for the encoder and decoder layers, bypassing the need for embedding layer distillation. The fine-tuning loss function is as

$$\mathcal{L}(\{\theta\}_{k=1}^K, \omega_S) = \sum_k^K \text{D}_{\text{KL}}(\sigma(f(\{\theta\}_k; \omega_k^f)) || \sigma(f(\{\theta\}_k; \omega_S^f))) . \tag{8}$$

Finally, starting with the model obtained after FedAvg, we fine-tune the model by minimizing the loss function as Eq (8). By employing several iterations of adversary sampling methods, we are able to gradually rectify the distributional discrepancies caused by FedAvg loss and enhance the performance of the model.

---

**Algorithm 1** FedDRS: Diversity Randomly Sample

---

**Input:** communication round $T$, client number $K$, the datasets of clients $\{\mathcal{D}\}_{k=1}^{K}$, parameters of student $\omega_S$, adversary sampling iterations $I$ and $I^*$, update steps $\eta, \eta^*$ and $\beta$.

**Output:** Global model parameters $\omega_S$.

1: **for** $t = 1 \rightarrow T$ **do**
2:     $\mathcal{S}_t \leftarrow$ select active clients uniformly at random
3:     **for** $k \in \mathcal{S}_t$ **do**
4:         $\omega_k \leftarrow \text{ClientUpdate}(\omega_S; D_k, \varsigma)$
5:     **end for**
6:     $\omega_S \leftarrow \text{FedDRS}(\{\omega\}_{k \in \mathcal{S}_t}, I)$
7: **end for**
8: $\omega_S \leftarrow \text{FedDRS}(\{\omega\}_{k=1}^{K}, I^*)$          ▷ Post-processing
9: **return** $\omega_S$
10: **FedDRS($\{\omega\}_{m=1}^{M}, I$):**
11:     $\omega_S \leftarrow \frac{1}{M} \sum_m^M \omega_m$
12:     **for** $i = 1 \rightarrow I$
13:         sample a proxy dataset $\{\theta^{rd}, \theta^{tr}, \theta^{ad}\}_{m=1}^{M}$, and pseudo labels $\{\gamma^{tr}, \gamma^{ad}\}_{m=1}^{M}$
14:         $\theta_k^{tr} \leftarrow \theta_k^{tr} - \beta \nabla_{\theta_k^{tr}} \mathcal{L}_{tar}$
15:         $\theta_k^{ad} \leftarrow \theta_k^{ad} - \beta \nabla_{\theta_k^{ad}} \mathcal{L}_{adv}$
16:         $\omega_S \leftarrow \omega_S - \eta^* \nabla_{\omega_S} \mathcal{L}(\{\theta^{rd}, \theta^{tr}, \theta^{ad}\}_{m=1}^{M}, \omega_S)$
17:     **end for**
18: **return** $\omega_S$

---

We have placed the pseudocode for the timing of sampling and distillation in Algorithm 1, and we summarize the detailed logic flow for sampling from Embeddings in Figure 2. As an expert in the field of Federated Learning and Knowledge Distillation, I have overseen the integration of these components to optimize model performance. During $T$ rounds of communication, the server selects a group of online trainable clients (often simulated using a random number in experiments). The global model is then sent to the clients for updates. After one round of communication, we start with an average parameter as the starting point for distillation. Through $I$ rounds of sampling and fine-tuning, we obtain the best model for that round. In the final round, we perform a post-processing step by increasing the parameters $I$ and the adversarial term $\lambda$, thereby enhancing the adversarial strength to achieve the best performance in the last round.

## 5 EXPERIMENTS

In this section, we commence by conducting comparative baseline experiments on the effects of FL with Cross-silo knowledge distillation on complex NLP understanding tasks SuperGLUE (Wang et al., 2019). Subsequently, we proceed with ablation experiments to investigate the individual effects of various sampling methods, parameters, and other components within the experimental setup. The SuperGLUE benchmark is a complex multi-task test set designed to evaluate the performance of natural language understanding models, challenging them with more difficult language tasks.

### 5.1 EXPERIMENT CONDITION

We considered various text classification tasks and chose the SuperGLUE benchmark to construct our experimental environment. The SuperGLUE benchmark (Wang et al., 2019) is a complex multi-task test set designed to evaluate the performance of natural language understanding models, challenging them with more difficult language tasks. It represents challenging NLP general tasks and is suitable for the properties of imbalance and non-iid in FL.

**SuperGLUE Benchmark**    The SuperGLUE Benchmark (Wang et al., 2019) is a natural language understanding (NLU) task evaluation benchmark designed to test and advance the ability of machines to comprehend natural language. It consists of a set of more challenging tasks that encompass semantic diversity, requiring models to possess higher levels of reasoning, inference, and contextual understanding. In comparison to some simpler NLU benchmarks, SuperGLUE places a greater em-

Figure 3: Dirichlet Distribution of Tasks on Clients. The figure depicts the allocation of training sets for various tasks in SuperGLUE, ranging from completely independent distributions to identical distributions with respect to the parameter $\alpha$, where $\alpha \to 0, \ 0.05, \ 0.5$.

phasis on evaluating the comprehensive understanding and generalization capabilities of models, providing researchers with a more rigorous testing environment to foster advancements in the field of natural language processing.

**Data distribution**    We use the Dirichlet distribution with a parameter $\alpha$ to create varying degrees of non-iid in our task dataset. In our work, we define the scenario where the parameter $\alpha$ of the Dirichlet distribution approaches zero. In this case, the distribution generates the identity matrix, allocating all samples of each category exclusively to a single client. As we increase the Dirichlet alpha parameter to $0.05$ and $0.5$, the data becomes more independently and identically distributed (i.i.d.), with a total of twenty clients considered. The specific distributions of eight tasks with different alpha are shown in Figure 3.

**Experiment hyperparameter settings**    In the experiment, the learning rate for the fixed update of the model is set to $\eta = 1e - 3$ and fine-tuning learning rate $\eta^* = 1e - 5$. The learning rate for adjusting the sampling is set to $\beta = 1e - 1$, and the update is performed for 100 iterations. The size of a batch pseudo samples $\theta$ is $64 \times 64 \times 768$. As a conservative measure during communication, we set adversary sampling iterations $I = 1, I^* = 3$. In the cross-silo scenario, we assume an $80\%$ participation rate for all clients. The parameter $\lambda$ controls the strength of the adversarial is always set by $0.1$. Each model is trained 5 times within the client, with an echo value of 5. In each training iteration, 250 samples are randomly selected from the client for training. Adafactor is used for updating all models. All tests were conducted with a fixed random seed of 42.

**Baselines**    We conducted a rigorous comparative analysis between the classical algorithm FedAvg and the latest knowledge distillation-based algorithms FedDF, FedKD, and FedAUX. For FedDF and FedAUX, we utilized the BookCorpus dataset as auxiliary data, extracting 16,000 random samples. The distillation process involved a step size of 1e-5 and 1 epoch of fine-tuning. To ensure a fair comparison, FedKD employed two equally sized RoBERTa models for local mutual distillation, without utilizing SVD during communication. The differential privacy component was excluded from FedAUX. Two representative Transformers were selected for the experiment: the classical RoBERTa (Liu et al., 2019) + MLP discriminative model (encoder only) and the T5-base text-to-text generation model (with encoder and decoder both). In the context of all SuperGLUE tasks, a model performs multiple tasks simultaneously. The specific data processing methods and labeling approaches for the SuperGLUE dataset have been detailed in Appendix A.

## 5.2    Main Experimental Analysis

The experimental results are shown in the Table 1. For the RoBERTa-Base model, our algorithm FedDRS utilizes a mixed sampling approach including all three sampling schemes, so-called Fed-DRS(mixed). FedDRS(mixed) exhibits a maximum improvement of up to 2 points and becomes the best algorithm in extremely unbalanced data distribution when $\alpha$ approaches 0. As the parameter $\alpha$ increases and the data distribution becomes close to iid, our FedDRS(mixed) still keeps at the top although the scores of baselines also increase. Overall, we conclude that FedDRS gets the best scores in this imbalanced and non-iid scenario, and it does not need auxiliary data like FedAUX or FedDF.

For the T5-base model, we take FedAvg as the only baseline. Because the lack of labeled auxiliary data as the inputs of the decoder part, it is challenging to conduct experiments using FedAUX and FedDF approaches. Due to the minor improvement that can be neglected in target sampling, we

| SuperGLUE | | Dirichlet | | |
|---|---|---|---|---|
| | | 0 | 0.05 | 0.5 |
| Model | Algorithms | C=8 | C=20 | |
| RoBERTa-Base | FedDRS(mixed) | **69.06±0.42** | **70.27±0.63** | **70.37±1.88** |
| | FedAUX | 67.07±0.29 | 69.93±1.01 | 70.37±0.75 |
| | FedDF | 66.55±1.25 | 67.11±0.96 | 69.40±0.78 |
| | FedAvg | 64.24±0.95 | 67.68±1.71 | 69.65±1.07 |
| | FedKD(2xRoBERTa) | 66.97±0.41 | 64.34±0.98 | 68.30±0.42 |
| T5-Base | FedDRS(AdOnly) | **72.95±0.95** | 70.82±0.85 | 72.50±0.51 |
| | FedDRS(PostOnly) | 71.36±0.00 | **72.70±0.02** | **72.70±0.01** |
| | FedAvg | 70.17±0.75 | 71.65±0.76 | 71.51±0.35 |

Table 1: SuperGLUE Dev Scores for FedDRS and baselines which presents the performance of two type of Transformers on three different data distributions using various FL algorithms for the last five rounds of communication, measured by the average score ± standard deviation.

only employed adversary sampling (AdOnly). For some cases such as $\alpha = 0.05$, the satisfactory performance of FedAvg achieving balanced updates and adversarial sampling did not improve effectively, we opted to perform post-processing only (PostOnly) based on FedAvg. This approach can not only enhance the model's effectiveness but also reduce computational costs. The Experiment results show that our algorithms also work best on T5-Base.

A series of experiments demonstrate that diversity sampling techniques are better for the non-iid distribution. FedDRS(mixed) can compensate for missing distributions to a greater extent. FedDRS(AdOnly) is a stable approach to enhancing model performance. Post-processing offers higher flexibility and can further enhance model performance. By effectively combining multiple strategies, it is possible to maximize model performance.

**Communication Cost Analysis** FedAUX involves transmitting an initial auxiliary dataset and score information matrix, resulting in higher communication overhead than FedAVG, which only requires local client updates during communication. The overall communication overhead of FedDRS and FedDF is comparable to FedAVG when considering only label distribution information, with negligible model costs. For FedKD, which employs smaller models for communication, its loss is expected to be smaller than FedAvg, but the model's efficiency is comparatively lower.

## 5.3 ABLATION STUDY

**The effects of different sampling methods** We compared the effects of three data generation schemes: random sampling, target sampling, and adversary target sampling, alongside their combinations, on the performance of FedAvg. By evaluating the first communication round with $\alpha$ approaches 0 and $I = 1$, we measured the improvements in model performance obtained from each sampling method. Each algorithm underwent 10 rounds of sampling and testing, with average scores calculated. The testing process was controlled using a fixed random seed to eliminate random value influences. Results in Table 2 showed improvements from each method. Combining the three methods produced diverse synthetic samples, and the hybrid algorithm yielded a performance enhancement compared to the sum of the individual effects of the method. Thus, the mixed samples enhanced the benefits of all three sampling techniques.

| Sample Method | Accuracy(%) | Improvement(%) |
|---|---|---|
| FedAvg | 34.95 | - |
| +random sample | 35.94 | 0.99 |
| +target sample | 35.54 | 0.59 |
| +adversary sample | 36.41 | 1.45 |
| FedAvg+MixSample | **38.25** | **3.30** |

Table 2: Accuracy of RoBERTa Improvement by Diversity Sampling Methods in the Initial Communication

**The performance of FedDRS on models without embeddings** In order to verify the applicability of FedDRS(PostOnly) to models lacking an Embedding layer, we conducted experiments under con-

| PostOnly | Method | Accuracy | RS | TS | AS | TA | RTA |
|---|---|---|---|---|---|---|---|
| CIFAR 100 (ResNet18) | FedAvg | 39.38 | -0.07 ↓ | 0.04 ↑ | 0.09 ↑ | 0.10 ↑ | 0.04 ↑ |
| | FedOpt | 54.42 | -0.05 ↓ | 0.00 | 0.02 ↑ | 0.03 ↑ | 0.02 ↑ |
| EMNISTCR(CNN) | FedAvg | 84.61 | 0.00 | 0.00 | 0.00 | 0.00 | 0.00 |
| | FedOpt | 84.88 | 0.00 | 0.00 | 0.00 | 0.00 | 0.00 |

Table 3: The enhancement effects of various sampling methods on top of other optimization techniques, including RS (Random Sampling), TS (Target Sampling), AS (Adversary Sampling), TA (TS&AS), and RTA (RS&TS&AS), are investigated.

ditions consistent with the TFF Benchmark (Reddi et al., 2020), with subsequent post-processing. As indicated by the results in Table 3, the overall effect exhibits a marginal improvement with only slight decreases. Random sampling often leads to reduced performance, but alternative sampling strategies yield only minor enhancements. This experiment illustrates that FedDRS is better suited for distillation with models that possess an embedding layer.

**Choices of adversarial functions and suitable adversarial strength** $I$    In adversarial sampling, we introduce an adversarial strength coefficient $I$ and compare the magnitudes of three different sampling strengths. Ultimately, we find that a strength of 0.1 can precisely yield a high-quality sample with a certain level of adversarial strength. We also evaluate the effect of replacing CE with KLD, as shown in the graph. The KLD curve abruptly decreases after a prolonged convergence, failing to produce a consistently high-quality sample. Therefore, CE outperforms KLD in terms of stability.

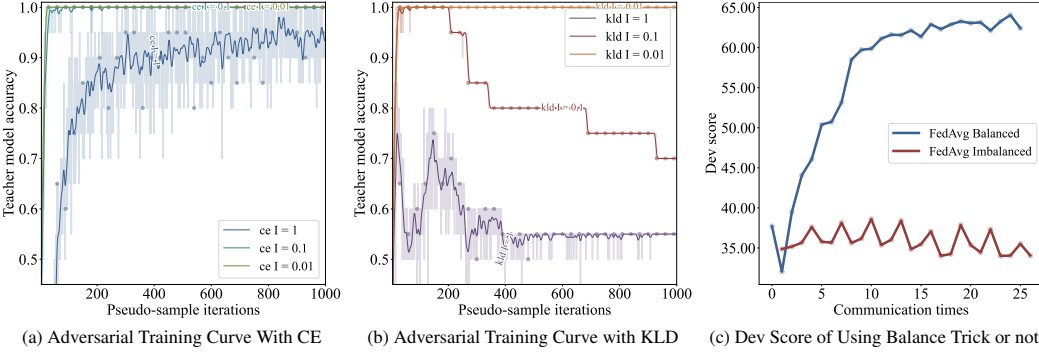

(a) Adversarial Training Curve With CE     (b) Adversarial Training Curve with KLD     (c) Dev Score of Using Balance Trick or not

Figure 4: The quality curve (a,b) of samples obtained during sampling iterations varies with the change in sampling iterations, and the KLD, as an adversarial term, fails to stably sample high-quality data points. In general, using CE with $I = 0.1$ ensures both adversarial strength and sample quality. (c) shows the dev score of the global model with communication times by using the balance trick or not. The balance trick on different clients makes the dev score much higher.

**Data balance trick**    It is worth noting that data is often highly imbalanced in a cross-silo setting. Aggregating models with sample quantity as weights can lead to severe unfairness. In our experiments, we employed a simple balancing trick by constraining the training of each client not to exceed a certain threshold. For SuperGLUE, we controlled the number of training samples for each client not to exceed the count of the client with the fewest samples at the current iteration.

## 6    CONCLUSION AND FUTURE WORK

To address the challenges of transformer distillation in Federated Learning involving GANs and auxiliary text, we propose three methods for sampling from the embedding layer. Across various complex tasks constructed within the FL-supergule framework, our approach outperforms methods that utilize auxiliary data. This approach is lightweight, incurring no additional communication overhead, and exhibits the most significant performance gains in non-iid scenarios. However, it is worth noting that sampling from the model still raises privacy concerns. In our future work, we intend to incorporate privacy-preserving measures, such as differential privacy, to ensure the privacy of the pseudo-samples.

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

# A   APPENDIX

## A.1   SUPERGLUE DATASET

The numbers of samples in each task of SuperGLUE are shown in Tabel.4.

| Corpus | Train/Dev(Test) | Cut Train | Task type |
|--------|-----------------|-----------|-----------|
| BoolQ | 9427/3270 | 9427 | QA |
| CB | 250/57 | 250 | NLI |
| COPA | 400/100 | 400 | QA |
| MultiRC | 5100/953 | 963 | QA |
| ReCoRD | 101k/10k | 9000 | QA |
| RTE | 2500/278 | 2500 | NLI |
| WiC | 6000/638 | 6000 | WSD |
| WSC | 554/104 | 554 | coref. |

Table 4: (Cutted) SuperGLUE Dataset (Wang et al., 2019). To prevent a single task type from dominating the allocation of all client cuts due to excessively large data volumes in our experiment settings, we appropriately trimmed the training data. QA is a question-and-answer task.

## A.2   PREPROCESSED EXAMPLES

In this section, we proposed our preprocessed examples of each task in the SuperGLUE dataset.

### A.2.1   BOOLQ

Original Input

> Question: science begins with the premise that knowledge should first be acquired through observation
>
> Passage: A priori and a posteriori – These terms are used with respect to reasoning (epistemology) to distinguish "necessary conclusions from first premises" (i.e., what must come before sense observation) from "conclusions based on sense observation" (which must follow it). Thus, the two kinds of knowledge, justification, or argument, may be glossed:

Processed Input

> boolq question: science begins with the premise that knowledge should first be acquired through observation. passage: A priori and a posteriori These terms are used with respect to reasoning epistemology to distinguish necessary conclusions from first premises ie what must come before sense observation from conclusions based on sense observation which must follow it Thus the two kinds of knowledge justification or argument may be glossed

Original Target: 0

Processed Target: Yes

### A.2.2   WIC

Original Input

> Word: place
>
> Sentence1: Do you want to come over to my place later?
>
> Sentence2: A political system with no place for the less prominent groups.

Processed Input

> wic word: place. sentence1: Do you want to come over to my place later. sentence2: A political system with no place for the less prominent groups

Original Target: 0

Processed Target: Mismatch

### A.2.3 WSC

Original Input

> Text: Mark told Pete many lies about himself, which Pete included in his book. He should have been more skeptical.
> span1_text: Mark
> span2_text: He

Processed Input

> wsc: Mark told Pete many lies about himself which Pete included in his book * He * should have been more skeptical

Original Target: 0

Processed Target: Difference

### A.2.4 CB

Original Input

> Premise: It was a complex language. Not written down but handed down. One might say it was peeled down.
> Hypothesis: the language was peeled down

Processed Input

> cb hypothesis: the language was peeled down. premise: It was a complex language Not written down but handed down One might say it was peeled down

Original Target: 0

Processed Target: Entailment

### A.2.5 RTE

Original Input

> Premise: No Weapons of Mass Destruction Found in Iraq Yet.
> Hypothesis: Weapons of Mass Destruction Found in Iraq.

Processed Input

> rte hypothesis: Weapons of Mass Destruction Found in Iraq. premise: No Weapons of Mass Destruction Found in Iraq Yet

Original Target: 0

Processed Target: Not_entailment

### A.2.6 ReCoRD

Original Input

> Passage: The harrowing stories of women and children locked up for so-called 'moral crimes' in Afghanistan's notorious female prison have been revealed after cameras were allowed inside. ... Crimes include leaving their husbands or refusing an arrange marriage 62 children live there and share cells with their mothers and five others
> Query: The baby she gave birth to is her husbands and he has even offered to have the courts set her free if she returns, but @placeholder has refused
> Entities: 'Mariam', 'Badam Bagh', 'Nuria', 'Afghanistan'

Processed Input

> record answer: Mariam. query: The baby she gave birth to is her husbands and he has even offered to have the courts set her free if she returns but @placeholder has refused. passage: The harrowing stories of women and children locked up for socalled moral crimes in Afghanistans notorious female prison have been revealed after cameras were allowed inside ... Crimes include leaving their husbands or refusing an arrange marriage 62 children live there and share cells with their mothers and five others

record answer: Badam Bagh. query: The baby she gave birth to is her husbands and he has even offered to have the courts set her free if she returns but @placeholder has refused. passage: The harrowing stories of women and children locked up for socalled moral crimes in Afghanistans notorious female prison have been revealed after cameras were allowed inside ... Crimes include leaving their husbands or refusing an arrange marriage 62 children live there and share cells with their mothers and five others

record answer: Nuria. query: The baby she gave birth to is her husbands and he has even offered to have the courts set her free if she returns but @placeholder has refused. passage: The harrowing stories of women and children locked up for socalled moral crimes in Afghanistans notorious female prison have been revealed after cameras were allowed inside ... Crimes include leaving their husbands or refusing an arrange marriage 62 children live there and share cells with their mothers and five others

record answer: Afghanistan. query: The baby she gave birth to is her husbands and he has even offered to have the courts set her free if she returns but @placeholder has refused. passage: The harrowing stories of women and children locked up for socalled moral crimes in Afghanistans notorious female prison have been revealed after cameras were allowed inside ... Crimes include leaving their husbands or refusing an arrange marriage 62 children live there and share cells with their mothers and five others

Original Target: Nuria

Processed Target: 'Wrong','Wrong','Correct','Wrong'

### A.2.7   COPA

Original Input

Premise: My body cast a shadow over the grass.
Choice1: The sun was rising.
Choice2: The grass was cut.
Question: Cause

Processed Input

copa choice1: The sun was rising. choice2: The grass was cut. premise: My body cast a shadow over the grass. question: cause

Original Target: 0

Processed Target: Choice_one

### A.2.8   MULTIRC

Original Input

Paragraph: While this process moved along, diplomacy continued its rounds. Direct pressure on the Taliban had proved unsuccessful. ... The U.S. effort continued.
Question: What did the high-level effort to persuade Pakistan include?
Answer: Children, Gerd, or Dorian Popa

Processed Input

multirc question: What did the highlevel effort to persuade Pakistan include? answer: Children Gerd or Dorian Popa. paragraph: While this process moved along diplomacy continued its rounds Direct pressure on the Taliban had proved unsuccessful ...The US effort continued

Original Target: 0

Processed Target: False

