# Text-Free Federated Transformers Knowledge Distillation Without GAN

## Abstract

Federated Learning (FL) is a distributed learning process designed to protect user privacy by avoiding the transmission of user data during communication while training a model. Many techniques aim to enhance the performance of models through knowledge distillation but lack data on the server side. To address this issue, Generative Adversarial Networks (GANs) are commonly employed to generate data for model distillation. The GANs approach faces numerous challenges in recent popular large-scale Transformer-based NLP tasks, such as structural mismatches in models, high computational complexity, and concerns regarding the privacy of client-generated text. Prior research has sought to enhance the process using auxiliary data to avoid the above issues, however, the selection of suitable data tailored to diverse tasks remains a challenging endeavor. To address the challenges posed by GANs and auxiliary data, this work proposes a lightweight approach that samples from the embedding structure of Transformers and learns a set of pseudo data for the distillation process, which draws inspiration from the concept of soft prompts. This lightweight approach does not require GANs or auxiliary data, incurs no communication overhead, and yields improved model performance with relatively lower computational costs on the server side. Our experiments yield superior results compared to methods that rely on auxiliary data on complex NLP tasks such as the SuperGLUE Benchmark.

## 1 Introduction

Federated Learning (FL) is a privacy-preserving distributed learning technique that has gained significant popularity. With the advancement of deep learning, the increasing demand for data by models has raised concerns about data privacy. Presently, over 90 countries have established privacy protection laws and policies (Li et al., 2021). FL finds applications in diverse fields such as Natural Language Processing (NLP) (Venkateswaran et al., 2022), Computer Vision (CV) (Lin et al., 2020), Industrial Artificial Intelligence (IAI) (Hao et al., 2019), and Medical Informatics (Xu et al., 2021). Leading AI companies like Google (Bonawitz et al., 2019), Apple (Paulik et al., 2021), and Meta (Stojkovic et al., 2022) are actively developing this technology to safeguard user privacy.

FL typically involves multiple clients participating in the training of a shared model. Based on the computational capabilities of participating clients, FL can be categorized into Cross-device (Karimireddy et al., 2021), common among low-capacity clients like smartphones and wearable devices, and Cross-silo (Huang et al., 2021), prevalent in large organizations, hospitals, and other entities with substantial computational resources. Generally, FL is approached as an optimization problem, although alternative paths involving knowledge distillation techniques also exist. This work focuses on the non-iid and imbalance distillation issues within the Cross-silo scenario, with communication limitations less pronounced in larger organizations.

Models like the Transformer (Vaswani et al., 2017), which combine pre-training tasks, have achieved remarkable success in the field of NLP. Prominent Transformer models include BERT (Bidirectional Encoder Representations from Transformers) (Devlin et al., 2018), T5 (Raffel et al., 2020), and GPT (Generative Pre-trained Transformer)(Alec et al., 2018). Notably, OpenAI's recently released Chat-GPT (OpenAI, 2023) has garnered exceptional attention in intelligent question answering and text generation. However, while FL's major baselines often focus on simple image classification tasks, there is limited in-depth research on Transformers under the FL paradigm. The distinctive struc-

ture and training methodology of Transformers set them apart from conventional neural networks, making conventional FL techniques unsuitable for their training processes.

Federated Learning can be conceptualized as a model ensemble process, which shares similarities with the principles of knowledge distillation. The integration of knowledge distillation with FL (Sattler et al., 2021; Lin et al., 2020) often seeks to enhance the overall performance of the global model. However, applying knowledge distillation to FL necessitates overcoming the challenge of transmitting data from clients to the server. Consequently, various GAN-based methods (Zhu et al., 2021; Zhang et al., 2022) for generating synthetic data have emerged in the FL context, with GANs learning to produce pseudo-samples aligning with the client distributions, forming the foundation for incorporating knowledge distillation techniques.

**Challange of GANs** However, crafting a GAN-based framework for text generation in the context of Transformers is a challenging endeavor due to its inherent sparsity and complexity (Brophy et al., 2023; Alvarez-Melis et al., 2022). GANs (Goodfellow et al., 2020) typically consist of a generator and a discriminator engaged in an adversarial game.

In frameworks like FedGEN (Zhu et al., 2021) and similar approaches, the traditional discriminator is replaced with client models, thereby facilitating the generation of samples specific to each client. However, FedGEN lacks a generalized approach, and there is no fixed paradigm for designing various generator structures tailored to different tasks. Besides, designing a deep generator that matches the depth of a Transformer model poses substantial computational and communication overhead.

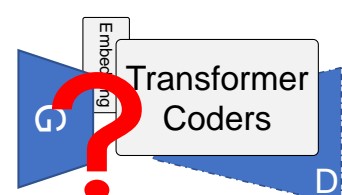

Figure 1: Creating an suitable and privacy-preserving generator for Transformers poses a formidable and intricate challenge.

Moreover, if one were to employ Transformers directly for generating client-side text sequences, privacy concerns arise. Research has shown that machine learning models can memorize data, allowing malicious actors to extract sensitive information from the model's behavior (Feldman & Zhang, 2020). As described in Guo et al. (2022), privacy attacks on pre-trained generative models include embedded inversion attacks, which can reverse engineer embedded code to infer the original sentences. Additionally, there are attribute inference attacks (Song & Raghunathan, 2020), where words or sentences from the training context exhibit more similarity scores compared to those from other contexts, thereby allowing inference attacks on the presence of certain words in the data. There are also corpus inference attacks (Carlini et al., 2021) and other attacks (Cai et al., 2021; Sundermeyer et al., 2012; Li et al., 2018) .

**Our contributions** In order to address the distillation challenge in FL, particularly in the context of Transformer models, especially when auxiliary data is scarce, and drawing inspiration from soft prompts, we propose a text-free approach that leverages diverse sampling from embeddings to effectively enhance model performance. Specifically, we design three methods for sampling from embeddings, with the core idea being to enhance distillation by sampling from embeddings and optimizing samples obtained through different objectives and their blends. This lightweight approach does not require GANs or auxiliary data, incurs no communication overhead, and yields improved model performance with relatively lower computational costs on the server side.

We conduct experiments on a variety of NLP understanding tasks from the SuperGLUE benchmark in a cross-silo FL setting, using two typical downstream task models (with or without decoder structures). Our results demonstrate superior performance compared to solutions relying on auxiliary data. Furthermore, our ablation experiments elucidate the unique advantages of models equipped with embeddings over those without embeddings, showcasing