# OpenReview forum: "Text-Free Federated Transformers Knowledge Distillation Without GAN"
_ICLR.cc/2024/Conference — ICLR 2024 Conference Withdrawn Submission_

### Official Review · Reviewer_PazT · 2023-10-25

**Soundness:** 2 fair
**Presentation:** 3 good
**Contribution:** 1 poor
**Rating:** 3
**Confidence:** 4

**Summary:**

The paper addresses the challenges in Federated Learning (FL) for NLP tasks, specifically the complications arising from using GANs and auxiliary data. By leveraging the embedding structure of Transformers, the authors propose a novel method to generate pseudo data inspired by soft prompts. This approach sidesteps the need for GANs, reduces computational overhead, and outperforms auxiliary data methods on the SuperGLUE Benchmark.

**Strengths:**

This paper has a clear presentation.

**Weaknesses:**

* **Motivation.** The motivation of this paper did not convince me. It seems that the target problem is ambiguous and meaningless. The authors seem to just make a minor modification to replace GAN in FL's knowledge distillation for NLP tasks and it lacks motivations and scenarios. GAN is actually rarely used in NLP and NLP is also less studied in FL before. Not using GAN in NLP is trivial and common, which cannot be the main motivation. An appropriate motivation is the problems raised in actual scenarios and previous works, not the "a + b" pattern. Also, the authors think GAN will leak privacy and the proposed method can protect privacy, but the authors didn't provide evidence to support that point.
* **Novelty.** I think the proposed method is not novel. First, knowledge distillation is not a novel thing in FL. Second, such a design in Transformers is also not novel.
* **Baselines.** The authors missed some important baselines in the experimental part, which weakens the validity of the proposed method. Specifically, the authors should compare the following methods in the experiments: [1] [2] [3].

----

[1] Zhang L, Shen L, Ding L, et al. Fine-tuning global model via data-free knowledge distillation for non-iid federated learning[C]//Proceedings of the IEEE/CVF conference on computer vision and pattern recognition. 2022: 10174-10183.

[2] Zhu Z, Hong J, Zhou J. Data-free knowledge distillation for heterogeneous federated learning[C]//International conference on machine learning. PMLR, 2021: 12878-12889.

[3] Wang H, Li Y, Xu W, et al. DaFKD: Domain-aware Federated Knowledge Distillation[C]//Proceedings of the IEEE/CVF Conference on Computer Vision and Pattern Recognition. 2023: 20412-20421.

**Questions:**

See the weakness above.

---

> ### Author Response · Authors · 2023-11-16
> **Motivation Clarification for Federated Learning with Non-GAN-based Text Distillation**
>
> We appreciate your time and expertise in reviewing my manuscript. Your constructive feedback has been instrumental in enhancing the quality of the paper.
>
> We acknowledge that our discussion regarding GANs and motivation has not been sufficiently elucidated. However, we persist in asserting the scarcity and necessity of research on non-text distillation in Federated Learning  within the domain of NLP. We possess compelling evidence that delineates the points of contribution we make to the field. Kindly afford us another opportunity to articulate our motivation more clearly in this context and present a more precise exposition.
>
> 1. Previous Data-free distillation of FL research (e.g., FedGEN, FedFTG) based on GAN often relies on image tasks (e.g., MNIST, CIFAR100) as primary benchmarks, neglecting NLP tasks. Other studies involving NLP tasks, such as FedAUX and FedDF, typically circumvent GANs and opt for auxiliary data distillation. Given the substantial gap in research on text tasks within FL's data-free distillation frameworks, exploring this area is both valuable and meaningful.
>
> 2. A practical issue arises with GAN in text generation [1](2023 NIPS). It is widely acknowledged that a notable obstacle in NLP GANs is their inability to generate differentiable outputs, given the discrete nature of language models (words). This lack of differentiability hampers mainstream Federated Learning (FL) Data-Free distillation frameworks, such as FedGEN[6] and FedFTG[7]. These frameworks utilize GANs for target learning functions but are ineffective in transmitting errors back to the generator, rendering them unsuitable for generating synthetic text in NLP distillation tasks under federated learning.
>
> 3. Can alternative text generation models be employed aside from GANs? Our response is that the proposed solution entails substantial costs in terms of privacy preservation and computational overhead.
>
> The current state-of-the-art in text generation, such as large-scale Transformer models like GPT-4, necessitates self-supervised training, involving memorizing client-side text data. However, this introduces privacy concerns, requiring intricate training mechanisms[2] and defense mechanisms[3] to ensure Transformer models remember the text while safeguarding privacy. Additionally, it raises the risk of attackers reconstructing text through pre-trained models[4,10], adding computational and communication overhead.
>
>
> - **Concerning privacy** complex training[2] and defense mechanisms[3] are required to ensure the Transformer remembers the text while protecting privacy. Simultaneously, precautions must be taken to prevent attackers from reconstructing text through pre-trained models[4,10], introducing both computational and communication overhead.
>
> - **Regarding computational overhead** deeper generative models (e.g., BERT, GPT) capable of memorizing more client-side text may exceed the computational capacity of the original task model, imposing a greater computational burden on clients.
>
>  - **Concerning communication overhead** the baseline Transformer for the original task already consumes substantial communication, such as FedKD[9] using shallow TinyBERT resulting in 0.17-1.03GB per mobile client. Introducing language generation models like FedGEN would further amplify communication overhead.
>
>
> In summary, GANs are unsuitable for text generation, rendering mainstream FL data-free distillation frameworks inadequate for text tasks. Instead of modifying the FedGEN framework, we propose a method that originates from the Transformer itself, generating data and distilling it. This approach aims to enhance model efficiency while circumventing the privacy, computational, and communication challenges associated with GANs. Compared to FedAUX and FedDF methods, our approach demonstrates superior performance in enhancing model efficiency.

---

> > ### Author Response · Authors · 2023-11-16
> > **Rebuttal:  Baselins , The authors missed some important baselines in the experimental part, which weakens the validity of the proposed method. Specifically, the authors should compare the following methods in the experiments: [1] [2] [3].**
> >
> > Due to our in-depth investigation of methods such as FedGEN, FedFTG, and DaFKD, we have uncovered the inapplicability of these approaches, which employ GANs on Transformer models in textual tasks.
> >
> > It is crucial to reiterate that our proposed method in the paper arises specifically due to the inadequacy of using GANs, as observed in FedGEN, FedFTG, and similar approaches, within the context of NLP tasks. Furthermore, it is noteworthy that even the reviewers acknowledge the infrequent utilization of GANs in NLP and the relatively limited exploration of NLP in the Federated Learning domain (GAN is actually rarely used in NLP and NLP is also less studied in FL before). In light of these considerations, we contend that these methods cannot serve as suitable baselines.
> >
> > Moreover, when considering the nature of the tasks, the fundamental tasks outlined in the three papers provided by the reviewer primarily pertain to image-related tasks. In contrast, our proposed method focuses on textual tasks. Therefore, we argue that these image-centric methods are not suitable as baselines for evaluating our approach.

---

> ### Author Response · Authors · 2023-11-16
> **References**
>
> [1] Alvarez-Melis, D., Garg, V., & Kalai, A. (2022). Are GANs overkill for NLP?. Advances in Neural Information Processing Systems, 35, 9072-9084.
>
> [2] Ponomareva, N., Bastings, J., & Vassilvitskii, S. (2022, May). Training text-to-text transformers with privacy guarantees. In Findings of the Association for Computational Linguistics: ACL 2022 (pp. 2182-2193).
>
> [3] Shangwei Guo, Chunlong Xie, Jiwei Li, L. Lyu, and Tianwei Zhang. Threats to pre-trained language models: Survey and taxonomy. ArXiv, abs/2202.06862, 2022
>
> [4] Zhang, R., Hidano, S., & Koushanfar, F. (2022). Text revealer: Private text reconstruction via model inversion attacks against transformers. arXiv preprint arXiv:2209.10505.
>
> [5] Song, C., & Raghunathan, A. (2020, October). Information leakage in embedding models. In Proceedings of the 2020 ACM SIGSAC conference on computer and communications security (pp. 377-390).
>
> [6] Zhang L, Shen L, Ding L, et al. Fine-tuning global model via data-free knowledge distillation for non-iid federated learning[C]//Proceedings of the IEEE/CVF conference on computer vision and pattern recognition. 2022: 10174-10183.
>
> [7] Zhu Z, Hong J, Zhou J. Data-free knowledge distillation for heterogeneous federated learning[C]//International conference on machine learning. PMLR, 2021: 12878-12889.
>
> [8] Wang H, Li Y, Xu W, et al. DaFKD: Domain-aware Federated Knowledge Distillation[C]//Proceedings of the IEEE/CVF Conference on Computer Vision and Pattern Recognition. 2023: 20412-20421.
>
> [9] Wu, C., Wu, F., Lyu, L., Huang, Y., & Xie, X. (2022). Communication-efficient federated learning via knowledge distillation. Nature communications, 13(1), 2032.
>
> [10] Carlini, N., Tramer, F., Wallace, E., Jagielski, M., Herbert-Voss, A., Lee, K., ... & Raffel, C. (2021). Extracting training data from large language models. In 30th USENIX Security Symposium (USENIX Security 21) (pp. 2633-2650).

---

> > ### Comment · Reviewer_PazT · 2023-11-20
> > **Post-rebuttal**
> >
> > Thanks for the authors' response. I have checked the rebuttal, but it seems it didn't convince me.
> >
> > Specifically, there may be no revision of the paper; even if it has, I didn't see the colored texts of change, making it hard to tell whether the authors can reclarify their contributions clearly. Also, no additional baseline experiments are added. The reason why the authors didn't add these baselines also didn't convince me.
> >
> > I appreciate the efforts they made during the rebuttal but I am sorry I cannot recommend acceptance currently. I wish this would not bother the authors and hope they can have a more solid version in the future.

---

> > > ### Author Response · Authors · 2023-11-22
> > > **Revision and Enhancement: Addressing Challenges in GAN-Based Data Generation for Transformer and SuperGLUE**
> > >
> > > Thank you for your suggestion! Originally, we intended to revise the motivation section of the paper, but unfortunately, our oversight resulted in a delay in making the necessary changes. We have now appropriately revised the manuscript and incorporated additional content in the updated version. We kindly ask you to reconsider giving us another chance.
> > >
> > > Regarding the issue you raised about baselines, using GANs (FedGEN, FedFTG) to generate data in Transformer and SuperGLUE is indeed a challenging task. To date, there is no existing literature addressing this aspect, primarily due to the high computational complexity involved. If there exists an algorithm capable of addressing this challenge, it would represent a completely new research paper, which diverges from our initial motivation. Once again, we appreciate your feedback, and we are committed to ensuring the thorough improvement of the paper.

---

### Official Review · Reviewer_hSV1 · 2023-10-29

**Soundness:** 3 good
**Presentation:** 3 good
**Contribution:** 3 good
**Rating:** 6
**Confidence:** 3

**Summary:**

This paper proposes a lightweight approach for knowledge distillation in federated learning (FL), particularly in the context of Transformer models. The authors address the challenges posed by Generative Adversarial Networks (GANs) and auxiliary data in FL by sampling from the embedding structure of Transformers and learning a set of pseudo data for the distillation process. This approach, called FedDRS, draws inspiration from the concept of soft prompts and does not require GANs or auxiliary data. It incurs no communication overhead and yields improved model performance with relatively lower computational costs on the server side.

The authors propose three methods for sampling from embeddings: random sampling, target sampling, and adversary sampling. They demonstrate that their approach outperforms methods relying on auxiliary data on complex NLP tasks such as the SuperGLUE Benchmark. The paper also presents ablation experiments that elucidate the unique advantages of models equipped with embeddings over those without embeddings, showcasing the efficiency and quality of sampling in embedding-enhanced models.

In summary, the paper introduces a novel text-free approach for knowledge distillation in federated learning, specifically for Transformer models. The proposed FedDRS method addresses the challenges posed by GANs and auxiliary data and yields improved model performance with lower computational costs.

**Strengths:**

### **Originality:**

The paper presents a novel approach for knowledge distillation in federated learning, particularly focusing on Transformer models. The proposed FedDRS method is unique in its text-free approach, which samples from the embedding structure of Transformers and learns pseudo data for the distillation process. This approach addresses the challenges posed by GANs and auxiliary data in FL, offering a creative combination of existing ideas.

### **Quality:**

The paper is well-written and provides a clear explanation of the proposed method. The authors demonstrate the effectiveness of FedDRS through experiments on the SuperGLUE benchmark, showing improved performance compared to methods relying on auxiliary data. The paper also includes ablation studies that elucidate the advantages of models equipped with embeddings.

### **Clarity:**

The paper is well-organized and presents its ideas in a clear and coherent manner. The authors provide a thorough explanation of the proposed method, its components, and the experimental setup. The results are presented in a clear and concise manner, making it easy for readers to understand the contributions of the paper.

### **Significance:**

The proposed FedDRS method addresses an important problem in federated learning, particularly in the context of Transformer models. By offering a lightweight approach that does not require GANs or auxiliary data, the method has the potential to advance the field of federated learning and improve the performance of Transformer models in FL settings. The paper also contributes to the understanding of the challenges posed by GANs and auxiliary data in FL, providing valuable insights for future research.

Overall, the paper presents a novel and creative approach to knowledge distillation in federated learning, focusing on Transformer models. The proposed FedDRS method demonstrates improved performance compared to existing methods and addresses the challenges posed by GANs and auxiliary data. The paper is well-written clear, and significantly contributes to the field of federated learning.

**Weaknesses:**

1. Privacy concerns (important): The paper does not address the potential privacy concerns arising from sampling from the model. Incorporating privacy-preserving measures, such as differential privacy, could help ensure the privacy of the pseudo-samples and enhance the overall robustness of the proposed method.

2. Limited exploration of sampling methods: The paper focuses on three sampling methods (random, target, and adversary sampling) but does not explore other potential sampling strategies. Investigating alternative sampling techniques could lead to further improvements in the performance of the proposed method.

3. Limited exploration of model architectures: The paper focuses on two Transformer models (RoBERTa and T5) but does not explore other popular Transformer architectures, such as BERT or GPT. Investigating the performance of the proposed method on a broader range of Transformer models could provide more insights into its applicability and effectiveness.

4. The illustration of Figure 1 seems chaotic.

**Questions:**

1. Although the authors mentioned about this weakness in the conclusion, it still requires some interpretation of how likely a generative model could leak private data. Therefore, I suggest authors add text inference attack experiments to show this risk.
2. In Table 3, I am curious about the performance of Fedavg + random sample + adv. sample. I suspect that the improvement of including a target sample in MixSample is negelactble.

---

> ### Author Response · Authors · 2023-11-16
> **Weakness Response.**
>
> Thank you very much for highlighting the weaknesses. Firstly, we acknowledge that privacy concerns constitute a weakness (and also a regret) in our work. Incorporating DP into Transformer is indeed a worthwhile research direction. We had considered integrating DP-SGD into the training process, but due to the inefficiency of per-sample gradients in the Transformer, we currently lack an ideal tool (such as JAX or Opacus) to address this issue, and thus, we had to abandon this approach.
>
> The three sampling methods are indeed limited, and in future work, we plan to introduce additional contrastive learning losses to explore the possibility of expanding sampling methods. Our focus is primarily on the two main training methods of Transformers, where RoBERTa represents self-supervised training, and T5 represents autoregressive training—both being mainstream and comprehensive paradigms. We did not consider structural diversity extensively because Transformers have several structural variants (BERT, GPT, T5, BART, etc.). The feasibility of incorporating experiments with other Transformer models will be discussed further.
>
> We are actively considering how to make the logic diagram clearer, such as reducing unnecessary process lines, and these modifications will be addressed in the next version of the paper. Once again, we appreciate your diligent work and valuable suggestions during the review process; they have positively influenced our research.
>
> Thank you once again to the reviewer. Your review has provided valuable guidance for our research. We will carefully consider the weaknesses and suggestions you pointed out and make the necessary revisions in the next version of the paper. Your professional insights have had a positive impact on our study, and we appreciate your efforts.

---

> ### Author Response · Authors · 2023-11-16
> **Questions Response**
>
> **Q:Although the authors mentioned about this weakness in the conclusion, it still requires some interpretation of how likely a generative model could leak private data. Therefore, I suggest authors add text inference attack experiments to show this risk.**
>
> A: Thank you for your suggestion. We believe that existing research can demonstrate this risk [1-4], and incorporating these studies, we intend to include the conclusions in the next version of the paper.
>
>
> [1] Ponomareva, N., Bastings, J., & Vassilvitskii, S. (2022, May). Training text-to-text transformers with privacy guarantees. In Findings of the Association for Computational Linguistics: ACL 2022 (pp. 2182-2193).
>
> [2] Shangwei Guo, Chunlong Xie, Jiwei Li, L. Lyu, and Tianwei Zhang. Threats to pre-trained language models: Survey and taxonomy. ArXiv, abs/2202.06862, 2022
>
> [3] Zhang, R., Hidano, S., & Koushanfar, F. (2022). Text revealer: Private text reconstruction via model inversion attacks against transformers. arXiv preprint arXiv:2209.10505.
>
> [4] Carlini, N., Tramer, F., Wallace, E., Jagielski, M., Herbert-Voss, A., Lee, K., ... & Raffel, C. (2021). Extracting training data from large language models. In 30th USENIX Security Symposium (USENIX Security 21) (pp. 2633-2650).
>
>
> **Q: In Table 3, I am curious about the performance of Fedavg + random sample + adv. sample. I suspect that the improvement of including a target sample in MixSample is negelactble.**
>
> A: We are pleased to conduct this experiment! Here are the experimental results (averaged over 10 runs):
>
> FedAvg + Random Sample + Target Sample = 36.8232138 (up 1.87)
>
> Based on these experimental results, it is evident that the contribution of the target sample is not negligible. We posit that the majority of the sample points obtained from target sample and adv sample sampling are not entirely redundant.
>
> ### Metrics for All Tasks:
>
> - AX-b  : 0.04466032370826787
> - AX-g  : 0.5120786516853932
> - BoolQ  : 0.3782874617737003
> - CB  : 0.034226190476190466
> - COPA  : 0.55
> - MultiRC  : 0.26065406168654304
> - ReCoRD  : 0.3278020452273851
> - RTE  : 0.5252707581227437
> - WiC  : 0.5042319749216301
> - WSC  : 0.36538461538461525
>
> **Scores** : 0.36823213844910097

---

> > ### Comment · Reviewer_hSV1 · 2023-11-22
> >
> > This is a great supplementary experiment, which has addressed my concerns.  For the rest of the weaknesses, the authors did not convince me, as they decided to leave the improvement further.

---

### Official Review · Reviewer_DJtk · 2023-10-31

**Soundness:** 2 fair
**Presentation:** 2 fair
**Contribution:** 2 fair
**Rating:** 5
**Confidence:** 4

**Summary:**

This paper proposes a lightweight approach for knowledge distillation in federated learning without using GANs or auxiliary data. The approach samples from the embedding structure of Transformers and learns a set of pseudo data for the distillation process, resulting in improved model performance with relatively lower computational cost. The paper suggests that this approach can be applied to other large-scale NLP tasks beyond Transformers.

**Strengths:**

* The approach does not require GANs or auxiliary data, incurs no communication overhead, and yields improved model performance with relatively lower computational costs on the server side.
* The experiments conducted in the paper show that the proposed approach yields superior results compared to methods that rely on auxiliary data on complex NLP tasks such as the SuperGLUE Benchmark.

**Weaknesses:**

* The challenge addressed in this paper may not be comprehensive. Although some papers utilize GANs to generate data for model distillation, it's important to note that GANs are not the sole method for data generation. Therefore, the scope of this paper appears to be limited.
* The assertion that "The GANs approach faces numerous challenges in recent popular large-scale Transformer-based NLP tasks" prompts the question: Were the models employed in the experiments considered large-scale?
* This paper does not specifically address the challenges associated with GAN-based methods for Federated Learning (FL) in its experimental section.
* Is this method applicable to other NLP tasks aside from text classification?

**Questions:**

See Weaknesses.

---

> ### Author Response · Authors · 2023-11-16
> **Motivation Clarification for Federated Learning with Non-GAN-based Text Distillation**
>
> Thank you for your valuable feedback and thoughtful review of our manuscript.
>
> We acknowledge that our discussion regarding GANs and motivation was not sufficiently clear. However, we persist in asserting the scarcity and necessity of research on non-text distillation in FL under NLP, and we present clear evidence supporting our contributions. Please grant us another opportunity to articulate our motivation more clearly and provide a more lucid argument:
>
> 1. Previous Data-free distillation of FL research (e.g., FedGEN, FedFTG) based on GAN often relies on image tasks (e.g., MNIST, CIFAR100) as primary benchmarks, neglecting NLP tasks. Other studies involving NLP tasks, such as FedAUX and FedDF, typically circumvent GANs and opt for auxiliary data distillation. Given the substantial gap in research on text tasks within FL's data-free distillation frameworks, exploring this area is both valuable and meaningful.
>
> 2. A practical issue arises with GAN in text generation [1](2023 NIPS). It is widely acknowledged that a notable obstacle in NLP GANs is their inability to generate differentiable outputs, given the discrete nature of language models (words). This lack of differentiability hampers mainstream Federated Learning (FL) Data-Free distillation frameworks, such as FedGEN[6] and FedFTG[7]. These frameworks utilize GANs for target learning functions but are ineffective in transmitting errors back to the generator, rendering them unsuitable for generating synthetic text in NLP distillation tasks under federated learning.
>
> 3. Can alternative text generation models other than GANs be employed? Our answer is: the solution's complexity is too high. The reasons are as follows:
>
> The current state-of-the-art text generation models, such as large Transformer models like GPT-4, exhibit the best performance. Using Transformer models for text generation necessitates training in a self-supervised manner, essentially requiring the model to memorize client-side text data.
>
> However, memorizing client-side text data raises the following issues:
>
> - Privacy concerns require intricate training mechanisms [2] and defense mechanisms [3] to ensure that the Transformer remembers this text while preserving privacy. Simultaneously, precautions must be taken to prevent attackers from reconstructing the text through pre-trained models [4]. This will increase both computation and communication overhead.
>
> - In terms of computation, the deeper the generative model (such as BERT or GPT-like self-supervised models), the more client-side text data it can remember. However, this may lead to computational overhead beyond the original task model, imposing a greater computational burden on clients.
>
> - In terms of communication overhead, the basemodel Transformer for the original task already consumes a substantial amount of communication, as seen in FedKD [9], where the use of a shallow model like TinyBERT incurs 0.17-1.03GB per mobile client. Incorporating language generation models, as in FedGEN, will further increase communication overhead.
>
> In summary, due to the escalated privacy, computational, and communication costs associated with incorporating text generation models into the FedGEN framework, we do not intend to address these issues. Instead, we propose an approach that originates from the Transformer itself, offering a method for distilling pseudo-data to Transformer models without the need for GANs. The effectiveness of this method surpasses that of using auxiliary data, as seen in FedAUX and FedDF, where the avoidance of GANs is also a deliberate choice.

---

> ### Author Response · Authors · 2023-11-16
> **Rebuttal : Other weakness**
>
> **•Q: The assertion that "The GANs approach faces numerous challenges in recent popular large-scale Transformer-based NLP tasks" prompts the question: Were the models employed in the experiments considered large-scale?**
>
> A:Taking reference from the papers on RoBERTa [9] and T5 [10-11], we utilized base models consistent with the original papers in our experimental section. The parameter sizes of these models are significantly larger, with RoBERTa having 102 million parameters [9], and T5 having 220 million parameters. These values far exceed the parameter sizes commonly used in federated learning, such as ResNet18 (11 million) and MLP, VGG, CNN (fewer parameters). Thus, we affirm that our experiments involve "large-scale" models.
>
> **•Q: Is this method applicable to other NLP tasks aside from text classification?**
>
> Yes, and not only limited to text classification tasks.
>
> In our paper, we regrettably did not explicitly provide detailed information about SuperGLUE, and we acknowledge this oversight. Thank you to the reviewer for pointing out this aspect.
>
> SuperGLUE encompasses a diverse set of text tasks beyond mere classification, including question answering, multiple-choice tasks, reading comprehension, and word sense disambiguation tasks [1-13]. It is a composite collection of tasks derived from various datasets, reflecting a combination of multiple tasks. The complexity of these tasks highlights the generality of our algorithm in addressing a wide range of natural language processing (NLP) challenges. For further details, please refer to: https://super.gluebenchmark.com/faq.

---

> > ### Author Response · Authors · 2023-11-16
> > **References**
> >
> > [1] Alvarez-Melis, D., Garg, V., & Kalai, A. (2022). Are GANs overkill for NLP?. Advances in Neural Information Processing Systems, 35, 9072-9084.
> >
> > [2] Ponomareva, N., Bastings, J., & Vassilvitskii, S. (2022, May). Training text-to-text transformers with privacy guarantees. In Findings of the Association for Computational Linguistics: ACL 2022 (pp. 2182-2193).
> >
> > [3] Shangwei Guo, Chunlong Xie, Jiwei Li, L. Lyu, and Tianwei Zhang. Threats to pre-trained language models: Survey and taxonomy. ArXiv, abs/2202.06862, 2022
> >
> > [4] Zhang, R., Hidano, S., & Koushanfar, F. (2022). Text revealer: Private text reconstruction via model inversion attacks against transformers. arXiv preprint arXiv:2209.10505.
> >
> > [5] Song, C., & Raghunathan, A. (2020, October). Information leakage in embedding models. In Proceedings of the 2020 ACM SIGSAC conference on computer and communications security (pp. 377-390).
> >
> > [6] Zhang L, Shen L, Ding L, et al. Fine-tuning global model via data-free knowledge distillation for non-iid federated learning[C]//Proceedings of the IEEE/CVF conference on computer vision and pattern recognition. 2022: 10174-10183.
> >
> > [7] Zhu Z, Hong J, Zhou J. Data-free knowledge distillation for heterogeneous federated learning[C]//International conference on machine learning. PMLR, 2021: 12878-12889.
> >
> > [8] Wang H, Li Y, Xu W, et al. DaFKD: Domain-aware Federated Knowledge Distillation[C]//Proceedings of the IEEE/CVF Conference on Computer Vision and Pattern Recognition. 2023: 20412-20421.
> >
> > [9] Liu, Y., Ott, M., Goyal, N., Du, J., Joshi, M., Chen, D., ... & Stoyanov, V. (2019). Roberta: A robustly optimized bert pretraining approach. arXiv preprint arXiv:1907.11692.
> >
> > [10] https://huggingface.co/t5-base.
> >
> > [11] Raffel, C., Shazeer, N., Roberts, A., Lee, K., Narang, S., Matena, M., ... & Liu, P. J. (2020). Exploring the limits of transfer learning with a unified text-to-text transformer. The Journal of Machine Learning Research, 21(1), 5485-5551.
> >
> > **SupeGLUE Benchmark Task Datasets References**
> >
> > [1] Roy Bar Haim, Ido Dagan, Bill Dolan, Lisa Ferro, Danilo Giampiccolo, BernardoMagnini, and Idan Szpektor. The second PASCAL recognising textual entailmentchallenge. 2006.
> >
> > [2] Luisa Bentivogli, Ido Dagan, Hoa Trang Dang, Danilo Giampiccolo, and BernardoMagnini. The fifth PASCAL recognizing textual entailment challenge. 2009.
> >
> > [3] Christopher Clark, Kenton Lee, Ming-Wei Chang, Tom Kwiatkowski, MichaelCollins, and Kristina Toutanova. BoolQ: Exploring the surprising difficulty ofnatural yes/no questions. In Proceedings of NAACL-HLT 2019, 2019.3
> >
> > [4] Ido Dagan, Oren Glickman, and Bernardo Magnini. The PASCAL recognisingtextual entailment challenge. In Machine learning challenges. evaluating predictiveuncertainty, visual object classification, and recognising tectual entailment, pages177–190. Springer, 2006.
> >
> > [5] Marie-Catherine De Marneffe, Mandy Simons, and Judith Tonhauser. The CommitmentBank:Investigating projection in naturally occurring discourse. 2019.To appear in proceedings of Sinn und Bedeutung 23. Data can be found athttps://github.com/mcdm/CommitmentBank/.
> >
> > [6] Danilo Giampiccolo, Bernardo Magnini, Ido Dagan, and Bill Dolan. The thirdPASCAL recognizing textual entailment challenge. In Proceedings of the ACLPASCALworkshop on textual entailment and paraphrasing, pages 1–9. Associationfor Computational Linguistics, 2007.
> >
> > [7] Daniel Khashabi, Snigdha Chaturvedi, Michael Roth, Shyam Upadhyay, and DanRoth. Looking beyond the surface: A challenge set for reading comprehensionover multiple sentences. In Proceedings of the 2018 Conference of the North AmericanChapter of the Association for Computational Linguistics: Human LanguageTechnologies, Volume 1 (Long Papers), pages 252–262, 2018.
> >
> > [8] Hector J Levesque, Ernest Davis, and Leora Morgenstern. The Winograd schemachallenge. In AAAI Spring Symposium: Logical Formalizations of CommonsenseReasoning, volume 46, page 47, 2011.
> >
> > [9] Mohammad Taher Pilehvar and Jose Camacho-Collados. WiC: The word-incontextdataset for evaluating context-sensitive meaning representations. In Proceedingsof NAACL-HLT, 2019.
> >
> > [10] Adam Poliak, Aparajita Haldar, Rachel Rudinger, J. Edward Hu, Ellie Pavlick,Aaron Steven White, and Benjamin Van Durme. Collecting diverse natural languageinference problems for sentence representation evaluation. In Proceedings ofEMNLP, 2018.
> >
> > [11] Melissa Roemmele, Cosmin Adrian Bejan, and Andrew S. Gordon. Choice ofplausible alternatives: An evaluation of commonsense causal reasoning. In 2011AAAI Spring Symposium Series, 2011.
> >
> > [12] Rachel Rudinger, Jason Naradowsky, Brian Leonard, and Benjamin Van Durme.Gender bias in coreference resolution. In Proceedings of NAACL-HLT, 2018.
> >
> > [13] Sheng Zhang, Xiaodong Liu, Jingjing Liu, Jianfeng Gao, Kevin Duh, and BenjaminVan Durme. ReCoRD: Bridging the gap between human and machine commonsensereading comprehension. arXiv preprint 1810.12885, 2018.

---

> > > ### Comment · Reviewer_DJtk · 2023-11-22
> > >
> > > some of my concerns have been addressed by the response of the authors, I would raise my score.

---

> > > > ### Author Response · Authors · 2023-11-22
> > > > **Think you for replying**
> > > >
> > > > Dear Reviewer,
> > > >
> > > > Thank you very much for thoroughly reviewing our manuscript and providing valuable feedback. We sincerely appreciate your critiques, which have given us the opportunity to enhance our research.
> > > >
> > > > We are delighted to see that you have increased the score for our work. This is both an encouraging and a learning opportunity for us. We are grateful for your recognition of our paper and will continue to strive for improvement.
> > > >
> > > > Sincerely,
> > > >
> > > > Paper Authors

---

### Official Review · Reviewer_PY1Q · 2023-11-06

**Soundness:** 3 good
**Presentation:** 3 good
**Contribution:** 2 fair
**Rating:** 5
**Confidence:** 4

**Summary:**

In this paper, the author propose a method to sample the embedding layer of transformer models and use for knowledge distillation in Federated Learning. The paper provides a good motivation to come-up with privacy preserving methods for knowledge distillation and identifies the gaps in GAN based methods.

**Strengths:**

This paper provides an interesting method to sample the embeddings of the transformer models for knowledge distillation in federated learning and thereby reducing the communication overhead and improving the accuracy.

**Weaknesses:**

The paper lack some important details about the proposed method and hence very difficult to read. In the abstract, it is mentioned, "This lightweight approach does not require GANs or auxiliary data, incurs no communication overhead, and yields improved model performance with relatively lower computational costs on the server side.". However, I don't see any discussion of the saving in communication cost later in the paper. Since the difference in accuracy is quite moderate as compared to FedAUX for various values of \alpha in Dirichlet distribution, we need to see what's the saving in communication cost and trade-off with additional computation cost at server.

Further, in the Ablation study, it's not clear that what numbers in Table 1 should be compared with the accuracy numbers given in Table 3.

Why do we see decaying performance difference between FedDRS and other techniques in Table 1 with increasing value of \alpha?

**Questions:**

please see above.

---

> ### Author Response · Authors · 2023-11-17
> **Discussion of the saving in communication cost**
>
> Thank you for your suggestions regarding communication overhead. The expression is not entirely clear to us. The absence of additional communication overhead refers to the comparison with the fundamental FedAVG approach, indicating the lack of additional communication costs.
>
> We will incorporate an analysis of this aspect in the experimental section of the next version of the paper, as outlined below:
>
> FedAUX necessitates the transmission of a substantial auxiliary dataset during the initial communication phase to compute the similarity between local and client datasets. Apart from model transmission, a score information matrix also needs to be communicated to clients in each communication round. This results in more communication overhead compared to FedAVG, encompassing both auxiliary data and similarity scores. Other non-data distillation solutions require the transmission of a generator based on FedAVG, thus adding to the communication overhead. FedAVG only requires the transmission of local client updates at time t during communication, and our algorithm's communication overhead is essentially equivalent to FedAVG (considering only label distribution information, with model costs being negligible).

---

> > ### Author Response · Authors · 2023-11-17
> > **Reply : Further, in the Ablation study, it's not clear that what numbers in Table 1 should be compared with the accuracy numbers given in Table 3.**
> >
> > We have noticed that we did not provide a detailed introduction to the SuperGLUE Benchmark, and we appreciate the reviewers for pointing this out.
> >
> > Tables 1 and 3 do not directly correspond to each other for comparison. Table 1 discusses how we constructed experimental data distributions conforming to the federated learning scenario. The metrics in Table 3 represent computed scores across multiple datasets in the SuperGLUE benchmark. SuperGLUE comprises various text tasks, including but not limited to question answering, multiple-choice, reading comprehension, and word sense disambiguation tasks [1-13]. It is a composite task set consisting of diverse tasks and datasets. The complexity of these tasks demonstrates the broad applicability of our algorithm to NLP tasks. For more information, please refer to https://super.gluebenchmark.com/faq.

---

> > > ### Author Response · Authors · 2023-11-17
> > > **Reply : 	Why do we see decaying performance difference between FedDRS and other techniques in Table 1 with increasing value of \alpha?**
> > >
> > > We need to elucidate the objective regularity of Federated Learning , namely that as $\alpha$ increases, the data distribution tends towards being independent and identically distributed (iid). However, in federated learning, a fundamental issue arises due to the non-iid nature of data, leading to a degradation in model performance.
> > >
> > > Many research endeavors in federated learning aim to rectify non-iid scenarios into iid scenarios. In such cases, the evaluation criterion for federated learning models is not about surpassing a specific algorithm but rather focuses on whether the performance can approach iid conditions when $\alpha$ is small.
> > >
> > > As $\alpha$ increases, the correctable gap diminishes, and the disparity between methods converges. Our FedDRS follows a similar pattern.

---

### Author Response · Authors · 2023-11-21
**Enhancements and Additions to Manuscript: Introduction Clarity, Contributions, and Benchmark Analysis**

Dear Esteemed Reviewers,

We have made revisions to the original manuscript, clearly delineating modifications in blue within the Introduction section, thereby enhancing the lucidity of our contribution. Additionally, we have incorporated elucidations regarding the SuperGLUE benchmark and provided an analysis of communication overhead in FedAUX.

---

### Meta-Review · Area_Chair_9hi4 · 2023-12-05

**Metareview:**

Federated Learning (FL) is often used to safeguard user privacy. This is achieved by circumventing the transmission of user data during communication while training a model. Numerous techniques strive to augment the performance of models via knowledge distillation, yet they often face the shortfall of data on the server side. To tackle this issue, Generative Adversarial Networks (GANs) are frequently utilized to generate data necessary for model distillation. The authors point out that the application of GANs, particularly in recent widespread large-scale Transformer-based NLP tasks, encounters several hurdles, including structural mismatches in models, elevated computational complexity, and privacy concerns related to client-generated text. To address these challenges, the authors propose an approach involving sampling from the embedding structure of Transformers and learning a set of pseudo data for the distillation process which is inspired by the concept of soft prompts. The authors claim that it does not necessitate the use of GANs or auxiliary data and avoids communication overhead. The authors carry out experiments to demonstrate that this approach yields superior outcomes relative to methods dependent on auxiliary data in NLP tasks such as the SuperGLUE Benchmark.

The reviewer for the most part found the method to sample the embeddings of the transformer models for knowledge distillation in federated learning interesting (although one reviewer had concerns about novelty). The reviewers did raise various concerns about lack of detail, the ablation study, lack of large-scale experiments, NLP tasks other than classification, privacy, limited exploration of sampling method and architectures, and concerns about novelty. While the authors response alleviated some of these concerns multiple reviewers thought their concerns were not fully addressed. I concur with the majority of the reviewers. The paper seems to have interesting insights but there are important concerns remaining (per reviewer discussions) that need to be addressed for publication. Therefore I can not recommend acceptance at this time.

**Justification For Why Not Higher Score:**

The average score is below 5, 3 out of 4 reviewers recommend rejection and the reviewers insist for the most part on their assessment post rebuttal.

**Justification For Why Not Lower Score:**

N/A

---

### Decision · Program_Chairs · 2024-01-16

Reject